# Echocardiography overestimates LV mass in the elderly as compared to cardiac CT

**Joshua Stokar[1], David Leibowitz[2], Ronen Durst[2], Dorith Shaham[3], Donna R. Zwas[ID][2]***

**1** Division of Endocrinology, Hadassah University Medical Center, Jerusalem, Israel, **2** Division of Cardiology, Hadassah University Medical Center, Jerusalem, Israel, **3** Division of Radiology, Hadassah University Medical Center, Jerusalem, Israel

* donnaz1818@gmail.com

**Data Availability Statement:** All relevant data are within the manuscript and its Supporting Information files.

**Funding:** The authors received no specific funding for this work.

## Abstract

### Purpose

Echocardiographic studies have shown an increase in LV mass with advanced age. However, autopsy and MRI studies demonstrate that with aging, LV mass is unchanged or slightly decreased, with a decrease in LV volume and an increase in wall thickness consistent with concentric remodeling. LV structural remodeling with aging may lead to an overestimation of LV mass in older adults when using standard echocardiography measurements and calculations. This study compared CT and echocardiographic LV mass calculation in younger and older patients and parameters associated with age-related LV remodeling.

### Methods

Same subject modality comparison of echocardiographic and cardiac CT LV measurement with derivation of LV mass was performed retrospectively. Echocardiographic measurements were performed by a single observer in accordance with European Association of Cardiovascular Imaging (EACI)/American Society of Echocardiography (ASE) guidelines. CT measurements were performed in end-diastole on multiplanar reformatted image planes corresponding to those typically used in echocardiography. Calculated CT measurements were based on automatic segmentation of heart chambers via edge-tracing algorithms.

### Results

129 patients were identified. In patients age 65 and older, LV mass was significantly higher when calculated using echocardiographic measurements compared to CT. Patients 65 years of age and older were found to have increased average wall thickness measurements with echocardiography but not with CT. The discrepancy between calculated echo and CT LV mass was reduced when using the mid-septal instead of proximal wall width for the EACI convention.

### Conclusion

In the elderly, increased echo-derived LV mass may reflect remodeling rather than a true increase in LV mass.

**Competing interests:** The authors have declared that no competing interests exist.

**Abbreviations:** CT, computerized tomography; LV, left ventricle; LVID, left ventricular internal diameter; LVM, left ventricular mass; LVOT, left ventricular outflow tract; MSWTd, mid-septal wall thickness; PWTd, inferolateral wall thickness; RWT, relative wall thickness; SWTd, septal wall thickness; LVV, left ventricular volume.

## Introduction

Echocardiographic assessment of left ventricular hypertrophy (LVH) predicts increased cardiovascular morbidity and mortality[1–3]. The increase in cardiovascular risk has shown to be continuously related to the degree of increase in LV mass[1], even within "normal range" values[4], and independent of hypertension and other risk factors [5]. LVH has been shown to be specifically associated with both systolic and diastolic dysfunction [6, 7]. Regression of previously increased LVM has been shown to be associated with a reduction in cardiovascular events and all-cause mortality [8]. The majority of studies evaluating the prognostic implications of LVM have utilized linear measurements of wall thickness and LV internal diameter, and subsequently calculated the volume of myocardium based on geometric assumptions of ventricular shape. As such, these calculations are subject to error both due to the assumptions of uniform ventricular shape, and as a result of the mathematical formula which may magnify small errors in measurement by a power of three. Despite these limitations, the robust epidemiologic prognostic data has led to widespread use of the echocardiographic assessment of LVH in clinical practice as a marker of cardiovascular prognosis.

CT and MRI-based measurement of LV mass utilizes edge-detection software to measure the inner and outer boundaries of the LV at different levels, and includes direct measurement of a 3-dimensional data set in the calculations, thus minimizing the geometric assumptions utilized in echocardiographic measurements. MRI based measurement is considered the "gold standard," and has been shown to be highly reproducible [9]. Validation of MRI measurement has been performed in-vivo in animal models and ex-vivo in human hearts [10–12]. CT based measurement has also been shown to be highly reproducible[13], and validated with necropsy with high correlation rates across a wide range of masses in both healthy and pathologically distorted hearts [13, 14].

Cross-sectional and longitudinal studies have found that calculated echocardiographic LV mass increases with advanced age, with up to 43% of otherwise healthy adults above age 70 considered to have LVH[15, 16]. At the same time, autopsy and MRI based studies have shown that the geriatric population has a normal or even decreased LV mass [17–21].

This discrepancy may be explained by the change in LV geometry seen in older age patients. Structural remodeling of the left ventricle is part of the normal aging process, even in the absence of disease. With aging, LV length decreases with an increase in wall thickness[17], and there is frequently discrete upper septal thickening [22] which leads to a decrease in total LV volume. These changes can increase the calculated echocardiographic mass because the septal wall width is increased and the ratio between LV length and short axis diameter is not maintained. This suggests that LV remodeling in older patients leads to overestimation of LV mass as per echocardiographic calculation when compared to CT or MRI based measurement. Thus, the accurate echocardiographic calculation of LV mass may require an adaptation of geometric assumptions as a function of age.

This study compared assessment of LV mass by CT and echocardiographic in patients 65 and older vs. younger patients.

## Methods

### Ethical approval

This study was approved by the Helsinki committee of Hadassah University Medical Center (0480-14-HMO) and was performed in accordance with the ethical standards of the institutional research committee and with the 1964 Helsinki declaration and its later amendments or

comparable ethical standards. The ethics committee waived the requirement for informed consent.

## Study design and participants

The study was a same-subject modality comparison study of patients aged 18 years and older who underwent a cardiac gated-CT scan between January 2010 and December 2013 and underwent transthoracic echocardiography at our institution within 6 months before or after the CT scan. Subjects with poor image quality were excluded from the study.

Demographic data were obtained from the computerized hospital patient management system.

## Echocardiographic measurements

Echocardiographic studies were performed on Vivid 7, Vivid 9, Vivid I or Vivid Q systems (GE) or IE33 (Philips). Echocardiographic measurements were performed off-line by a single observer in accordance with ASE/EACI guidelines[1] using the EchoPac system (GE Healthcare, Little Chalfont, Buckinghamshire, UK).

Measurements made in the parasternal long axis view included: septal wall thickness (SWTd), inferolateral wall thickness (PWTd), and left ventricular internal dimension (LVID.) Mid septal wall thickness (MSWTd) was measured in the mid ventricle at the level of the papillary muscles. (See Fig 1 for comparison to standard measurement.) Left ventricular outflow tract angulation was measured at the interception of a line drawn through the mid aorta and a mid-ventricular line from the mitral annulus to the apex (S1 Fig). Aortoseptal angulation was measured at the interception of a line parallel to the aortic valve and proximal septum with a drawn line parallel to the distal mid septum (S1 Fig). Area of the LV cavity (LVA) was measured at the level of the papillary muscles. LV length was measured from the level of the mitral annulus to the apex.

Computed measurements included:

$$SI = \frac{LV\ length}{radius \times 2}$$

$$Proximal : mid\ septum\ ratio = SWTd/MSWTd$$

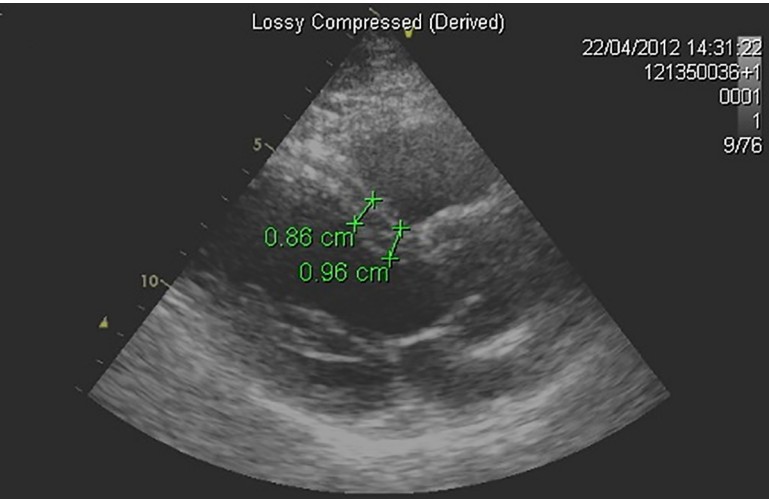

**Fig 1. Echocardiographic parasternal long axis view.** Echocardiographic parasternal long axis view–proximal (0.96cm) and mid (0.86cm) septal wall measurements.

$$Relative\ wall\ thickness\ (RWT) = \frac{SWTd + PWTd}{LVID}$$

$$LV\ volume = 5/6(LVA * LV\ length)$$

LV mass was calculated using the European Association of Cardiovascular Imaging linear cube formula:[23]

## CT measurements

Cardiac CT was performed on the Philips 256 Brilliance Scanner (Andover, MA) using prospective gating volumetric acquisition during diastole. CT measurements were performed in end-diastole on multiplanar reformatted (MPR) image planes corresponding to those typically used in echocardiography[23]. One reader performed all measurements. The reader was blinded to the echocardiographic measurements. Measurements included apical length in four chamber view, left ventricular internal diameters (4, 3, 2 chamber views), septal wall thickness, mid septal wall thickness, lateral wall thickness, and posterior wall thickness (S1 Fig and S2 Fig).

Left ventricular out flow tract angulation was measured at the intersection of a line parallel to the aorta and a line from the mitral annulus to the apex (S1 Fig).

Aortoseptal angulation was measured at the intersection between a line parallel to the aorta and a line parallel to the septal wall (S1 Fig).

Calculated CT measurements were based on automatic segmentation of heart chambers via edge-tracing algorithms with manual correction when necessary, utilizing the Philips Intellispace Portal Comprehensive Cardiac Package v4.02.51006 (2012) (Fig 2).

The volume of each heart segment was calculated by counting the overall pixels of the segment. The mass was then calculated by multiplying volume by myocardial density.

## Statistical analyses

Continuous variables are presented as mean ± standard deviation. Student's t-test was used to compare differences between two groups. Pearson's correlation was used to determine the correlation between measures. Systematic error and the degree of agreement were assessed with Bland Altman's analysis[24], and presented graphically using mountain plots (folded empirical cumulative distribution plots).[25]. A 2-tailed p value < 0.05 was considered to indicate statistical significance for all tests. All data were saved in Microsoft Excel 2013 spread sheets and analyzed using MedCalc 12.7.1 64-bit software for Microsoft Windows and IBM, SPSS Statistics for Windows, Version 22.0 (IBM, Armonk, NY).

## Results

129 patients were retrospectively identified and included in the study. Patient age ranged from 18 to 90, with a mean age of 59 ±17. 64% of the subjects were men, of whom 35% were 65 and older. 55% of the women were 65 and older. Median number of days between CT and echo studies was 3 with an interquartile range of 34 days.

## LV measurements: Echo vs. CT

Comparison of basic LV measurements on echocardiography and CT was notable for an average proximal:mid septal ratio that was on average 11% higher (p = 0.0001) on echocardiography

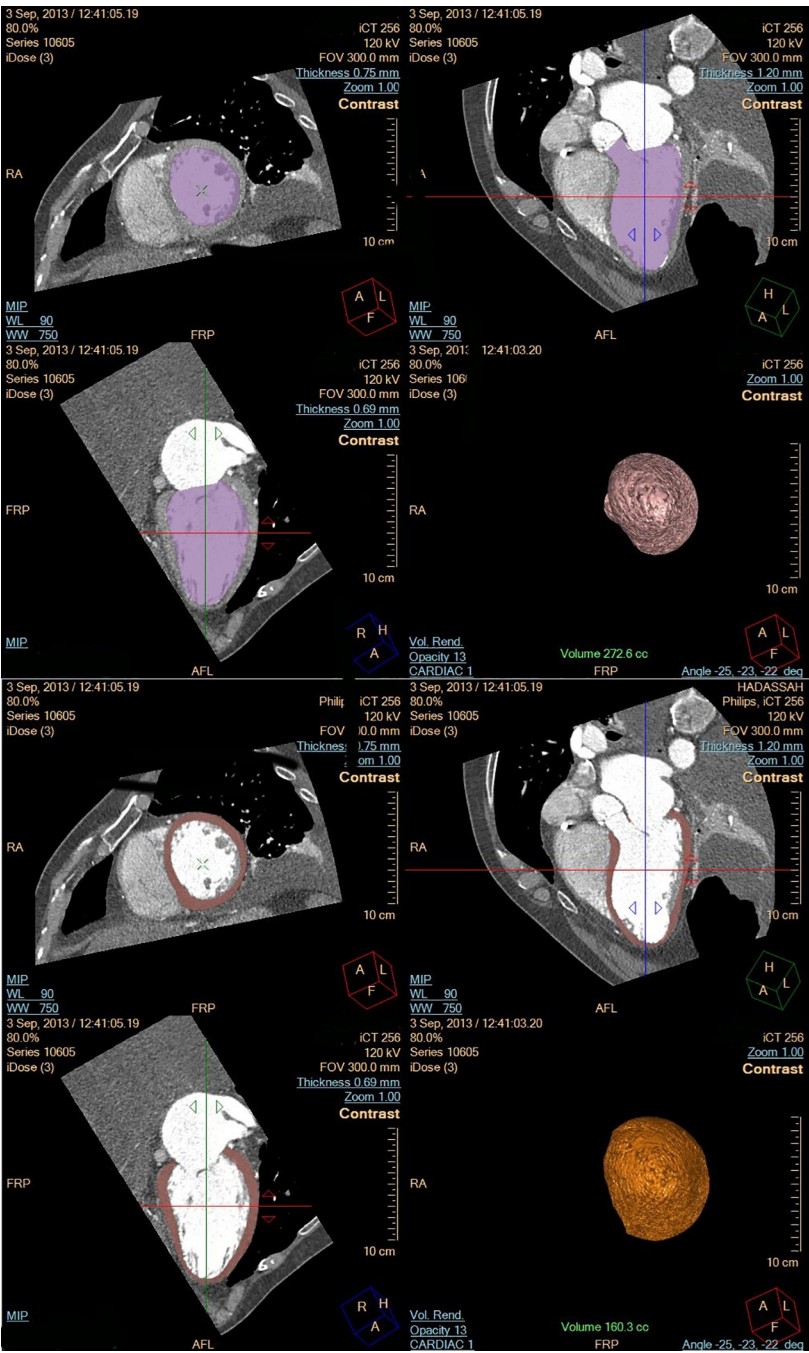

**Fig 2. CT–Determination of internal volume and LV myocardial volume.**

compared to CT. Additionally, average LV length was 8% higher (0.6cm, p<0.0001) on CT when compared to echo (Table 1). There was no significant difference between LV volumes as calculated by echocardiography and CT.

Average LV mass was found to be 12% higher (174 vs 155g) when calculated using echocardiographic measurements compared to the CT edge-detection based measurement (P = 0.0043).

**Table 1. Echocardiographic VS. CT measurements.**

|  | Echo | CT | P value |
|---|---|---|---|
| **SWT mid $_{cm}$** | 0.89 (0.86 to 0.92) | 0.97 (0.84 to 1.11) | 0.20 |
| **SWT $_{cm}$** | 1.02 (0.97 to 1.06) | 1.15 (1.11 to 1.12) | 0.48 |
| **RWT** | 0.39 (0.37 to 0.41) | 0.39 (0.35 to 0.42) | 0.9656 |
| **PWT $_{cm}$** | 0.94 (0.91 to 0.97) | 0.92 (0.81 to 1.04) | 0.79 |
| **Sphericity index** | 1.73 (1.68 to 1.78) | 1.86 (1.81 to 1.91) | 0.0004 |
| **LVID $_{cm}$** | 4.84 (4.72 to 4.96) | 4.83 (4.67 to 4.99) | 0.89 |
| **LV length $_{cm}$** | 8.42 (8.26 to 8.59) | 9.06 (8.86 to 9.26) | <0.0001 |
| **LVOT˚** | 141.15˚ | 132.58˚ | 0.119 |
| **Aorto-septal˚** | 138.04˚ | 128.55˚ | 0.37 |
| **LV volume $_{cm}{}^3$** | 140.74 | 143.48 | 0.5538 |
| **LV mass $_g$** | 174.46 (165.10 to 183.82) | 155.40 (146.23 to 164.58) | 0.0043 |

SWT = Septal wall thickness; RWT = relative wall thickness; PWT = posterior wall thickness; LVID = Left ventricular internal diameter; LVOT˚ = left ventricular outflow tract angle; Aorto-septal˚ = aorto septal angle

## Aging and LV measurements

A comparison between the older (age 65 years and above) and younger (age less than 65) patient group measurements using echocardiography and CT is presented in Table 2.

Several significant differences were found on imaging between the older and the younger groups. When comparing the older to the younger group, there was a greater increase in average proximal septal wall thickness measurements on echocardiography (25%:2.3mm)

**Table 2. LV measurements by age groups and correlation with age (ECHO VS CT).**

|  | Echo | | | | CT | | | |
|---|---|---|---|---|---|---|---|---|
|  | <65 | >65 | P value | R with age (95% CI) | < 65 | > 65 | P for difference | R with age (95% CI) |
| **SWT$_{cm}$** | 0.92 | 1.15 | <0.0001 | 0.46 (0.31 to 0.58) | 1.11 | 1.21 | 0.0185 | 0.28 (0.11 to 0.43) |
| **SWT mid$_{cm}$** | 0.85 | 0.93 | 0.0083 | 0.19 (0.02 to 0.35) | 1.03 | 0.90 | 0.2879 | -0.10 (-0.27 to 0.07) |
| **PWT$_{cm}$** | 0.91 | 0.98 | 0.0250 | 0.15 (-0.02 to 0.31) | 0.97 | 0.87 | 0.3228 | -0.07 (-0.24 to 0.10) |
| **LV length$_{cm}$** | 8.60 | 8.17 | 0.0088 | -0.28 (-0.44 to -0.11) | 9.30 | 8.74 | 0.0052 | -0.32 (-0.47 to -0.16) |
| **LVOT˚** | 142.86˚ | 138.78˚ | 0.0055 | -0.34 (-0.49 to -0.18) | 132˚ | 133˚ | 0.2835 | -0.11 (-0.28 to 0.06) |
| **Aorto Septal˚** | 143.86˚ | 130.02˚ | <0.0001 | -0.42 (-0.56 to -0.27) | 132˚ | 124˚ | 0.0067 | -0.32 (-0.46 to 0.15) |
| **RWT** | 0.37 | 0.41 | 0.0186 | 0.17 (0.01 to 0.34) | 0.40 | 0.37 | 0.2982 | -0.08 (-0.25 to 0.10) |
| **SI** | 1.75 | 1.70 | 0.3535 | -0.08 (-0.26 to 0.10) | 1.89 | 1.81 | 0.1013 | -0.12 (-0.29 to 0.05) |
| **LVID$_{cm}$** | 4.92 | 4.74 | 0.1613 | -0.09 (-0.26 to 0.08) | 5.01 | 4.90 | 0.45 | -0.12 (-0.29 to 0.05) |
| **LVV$_{cm}{}^3$** | 144.18 | 135.93 | 0.934 | -0.1514 (-0.33 to 0.03) | 147.16 | 137.69 | 0.577 | -0.13 (-0.29 to 0.046) |

SWT = septal wall thickness; PWT = inferolateral wall thickness; LVOT˚ = left ventricular outflow tract angle; Aorto-Septal˚ = aorto-septal angle; RWT = relative wall thickness; SI = sphericity index; LVID = left ventricular internal diameter; LVV = Left ventricular volume

compared to CT (9.5%:1mm). Estimates of average LV length were found to be 5% (echo) and 7% (CT) shorter in the older age group compared to the younger patients. The aortoseptal angulation was decreased in the older age groups compared to the younger group on both echo and CT.

Statistically significant increases were seen on echo when comparing average septal mid wall thickness (0.8mm) and inferolateral wall thickness (0.7mm) between older and younger patients, whereas non-significant decreases on mid septal and inferolateral wall thickness were seen on CT. No differences between age groups in either modality were seen in left ventricular internal diameter, volume or sphericity index.

### LV mass by age

A comparison of calculated LV mass values for the different age groups as derived from echocardiography measurements and CT is presented in Table 3. As mentioned above, average echocardiographic LV mass was significantly higher than CT derived mass. When stratified by age, a significant difference between modalities was found only in the older age group (189 vs. 165g, p<0.0048). Relative correlation coefficient between ASE and CT derived LV mass was 0.89; correlations by age group are presented in the scatter plot in Fig 3.

Absolute deviation of ASE LV mass from CT derived LV mass (19 ±52g) by age group is presented using Bland-Altman analysis in Fig 4.

Average ASE LV mass increased with age in women (r = 0.48, 95% CI 0.23 to 0.68, P = 0.0076) as well as men (r = 0.29, 95% CI 0.081 to 0.48, P = 0.0076). When using CT derived LV mass, no such correlation was found (r = -0.008, p = 0.9238) in either gender.

### Proximal:mid septal ratio

Increased proximal septal wall thickness in the older age group is seen on both echo and CT, leading to a higher proximal:mid septal wall ratio (1.25 on echo vs 1.07 on CT, p < 0.0001) which was found to be positively correlated with age (R = 0.42). A negative correlation was found between the proximal:mid septal wall ratio and the LV length per CT (R = -0.306, P = 0.0004); as well as with the aortoseptal angulation (R = -0.40, P<0.0001). A strong positive correlation was found between the proximal:mid septal wall ratio and the difference between echo and CT derived LV mass (R = 0.70, 0.60 to 0.78, P<0.0001).

### Echo calculation of LV mass: proximal vs. mid-septum

When using the mid-septal instead of proximal wall width for the ASE formula, the average LV mass was significantly lower for the older age group (165 vs. 189g, P = 0.0318) but remained relatively unchanged for the younger group (163 vs 156g, P = 0.2934). Accordingly, the positive correlation between LV mass and age disappeared when calculations were performed based on the mid-septal wall thickness (r = 0.06963, P = 0.4330). Using the mid-wall

**Table 3. Comparison of LV mass calculation methods.**

| Group (N) | ASE LV mass g (95% CI) | CT LV mass g (95% CI) | P value |
|---|---|---|---|
| All (129) | 174.46 (165.10 to 183.82) | 155.40 (146.23 to 164.58) | 0.0043 |
| ≥65 yrs (55) | 189.49 (173.17 to 205.80) | 156.50 (140.35 to 172.64) | 0.0048 |
| <65 yrs (74) | 163.29 (152.75 to 173.84) | 154.59 (143.66 to 165.52) | 0.2550 |

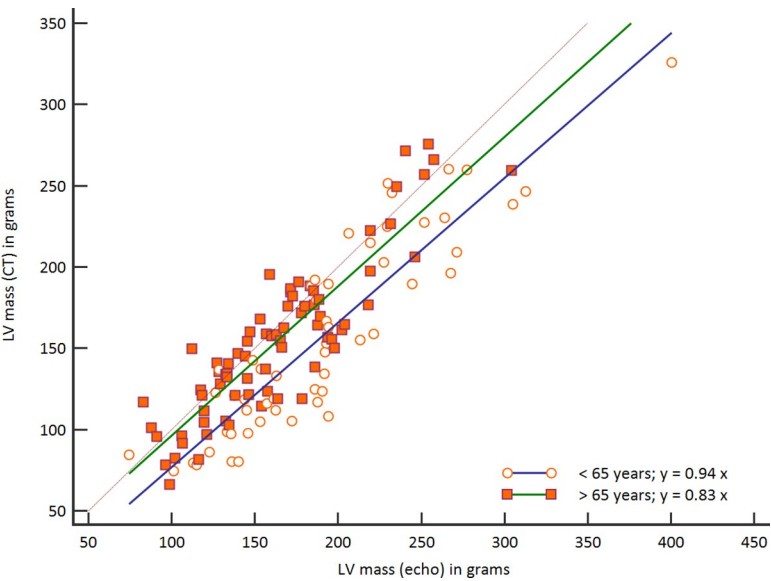

**Fig 3. Correlation between echocardiographic (ASE) and CT derived LV mass by age group.** Correlation between echocardiographic (ASE) and CT derived LV mass by age group. Brown line is line of equality.

thickness also eliminated the significant difference between LV mass on echo and on CT (159.7 vs 155.4 g, P = 0.4985).

Mountain plots for both age groups comparing the differences between CT and echocardiographic LV mass using proximal vs. mid septum measurements are shown in Fig 5.

Bland-Altman plots startified by age group comparing CT and echocardiographic LV mass measurement using proximal and mid septum are presented in Fig 4.

A similar correction of LV mass in the older age group was achieved when applying the proximal:mid septal ratio as a correction factor onto mass calculated from the ASE formula.

## Discussion

This study demonstrates a discrepancy between echo and CT calculations of LV mass, primarily in patients age 65 and older. Our findings using CT data are similar to those of a large MR based study, which also demonstrated no increase in LV mass with older age [20]. In that study, LV mass decreased incrementally by age, with a corresponding increase in mass to volume ratio mediated by a decrease in LV volume. In our study we did not see a statistically significant difference in LV volume, but we were underpowered to do so.

### LV mass and age

When examining other LV measures for their correlation with age, this study found a decrease in LV length, increased thickening of the proximal septum with a lesser degree of thickening of the posterior wall. The LV was found to shorten with prominent thickening of the proximal septum with advancing age. This buckling phenomenon, known as discrete upper septal hypertrophy, also known as a septal bulge, has been well-described in the elderly, and is found in up to 10% of large echocardiographic cohorts.[26, 27] This finding is associated with hypertension and diastolic dysfunction, but not with increased cardiovascular mortality.[28] It has been associated with increased LV mass in echocardiographic studies.[29] The septal bulge was quantified in this study using the proximal:mid septum ratio and was associated with

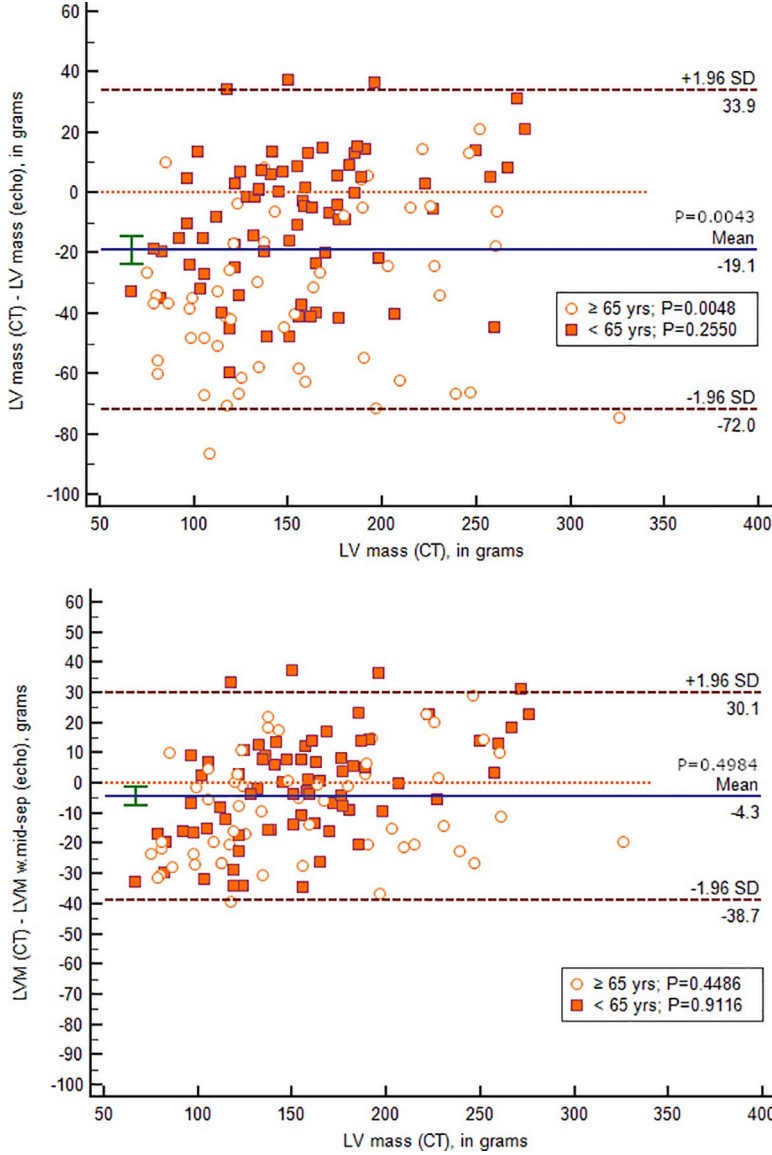

**Fig 4. Bland-Altman plots comparing LV mass measurement on echo and CT.** Bland-Altman plots comparing LV mass measurement on echo and CT. The upper plot uses proximal septal measurements and the lower plot uses mid-septal measurements. Both plots present sub-groups by age. P values for mean difference from 0 are presented for total and by age group.

differences in aortoseptal angulation and correlated with age. These findings were seen on both CT and echocardiography. As in previous echocardiographic studies, relative wall thickness was found to increase with age, a finding not seen in our study on CT.

As the ASE formula assumes a universal prolate ellipse shape of the LV, the formula is unlikely to take into account the septal bulge associated with aging. The original studies used to determine the equations for echo derived LV mass [27, 30, 31], consisted of relatively small numbers of subjects (34, 52, and 21 subjects), with a wide age range (23 to 82 years) and large variation in LV mass (77 to 625g). Given the observed changes in LV geometry with aging and the discrepancy with CT measured mass on this study, and the discrepancy with studies of

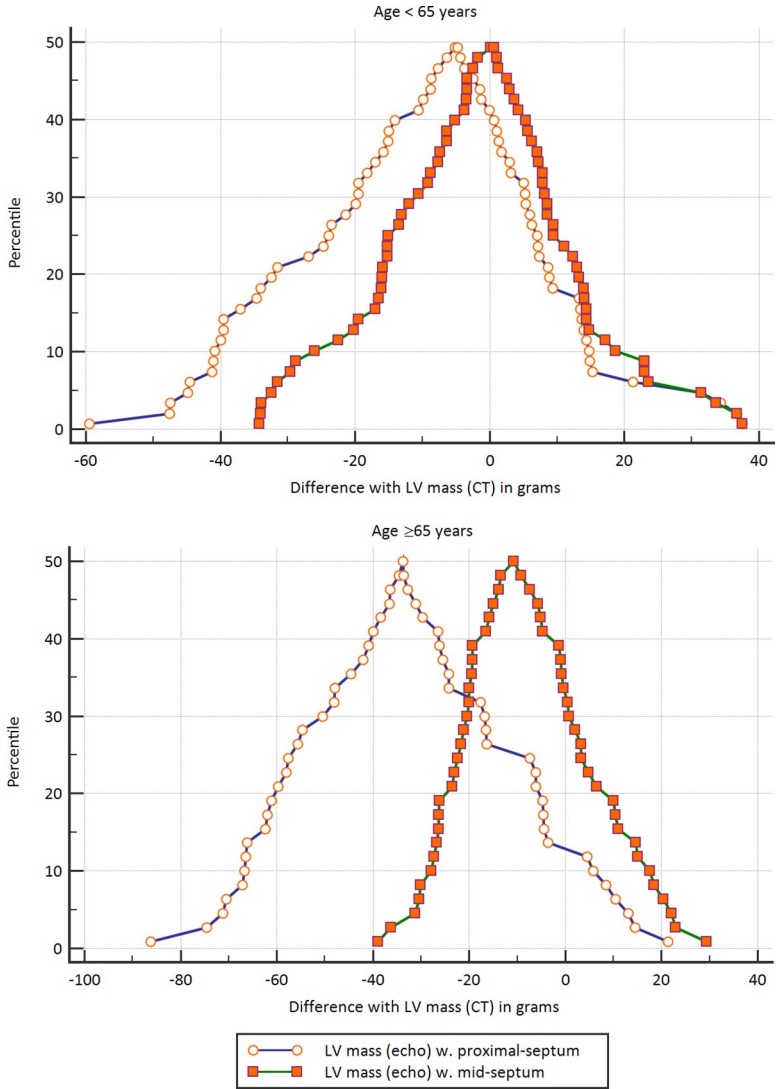

**Fig 5. Mountain plots.** Mountain plots (folded empirical cumulative distribution plots) presenting the percentile for each difference between the echo calculation and the CT calculation. Percentiles above 50 are presented as 100-percentile to achieve the folded plot. The upper plot demonstrates the difference between echo calculations using the proximal and mid septum in younger patients, and the lower plot presents the differences in patients 65 and older.

cardiac MR, some adjustment to the previously described echocardiographic formula may be warranted in older individuals.

## Proximal vs. mid septal measurement

When mid rather than proximal septal wall thickness was used in the ASE formula much of the echo-CT discrepancy was eliminated. The change in measurement region was insignificant for the younger age group, since the septum is usually the same width throughout in healthy young hearts, but it did make a difference in the presence of subtle thickening of the proximal septum, or in the presence of a discrete upper septal thickening. Clinical laboratories frequently measure distal to the septal knuckle, though the measurements may be done more proximally than what was performed in this study. As the proximal septal thickening is not

necessarily focal but may have a somewhat gradual attenuation, this study suggests that even measurements immediately distal to the septal bulge may lead to overestimation of LV mass. Although mid-ventricular measurement did not specifically adjust for the geometric changes, the estimate of LV mass correlated better with the 3-dimensional assessment performed by CT. This study suggests that mid-septal rather than standard proximal septal measurement may increase the accuracy of echocardiographic assessment of LV mass, as the septal bulge reflects remodeling and compression rather than an increase in overall ventricular mass.

This study's findings in no way minimize the well-established predictive power of echo-derived LV mass for cardiovascular prognosis and its positive correlation with age. Discrete upper septal wall thickening has been described as a risk factor even without LVH, though not independent of other risk factors like hypertension[22]. The prognostic epidemiological data obtained in large prospective studies of echo-derived LV mass is highly significant. It is possible that echo-derived LVH is a marker of LV remodeling and a reflection of the changes in LV geometry rather than a reflection of an increase in LV mass or relative wall thickness. The cardiac remodeling associated with aging may represent the loss of myocytes [32, 33], and increased fibrosis is found in animal and human autopsy studies [34]. Cardiac collagen content has been found to increase by close to 50% between the third and seventh decades of life [35]. Alternately, geometric changes of the aorta associated with decreased vascular distensibility may lead to conformational changes in the heart [36].

With further exploration of LV remodeling with echo and other imaging studies, it may be possible to derive other more accurate descriptors of LV remodeling that improve the prognostic capabilities of what is now described as echo-derived LV mass.

Our study does have several limitations. MRI is considered the gold standard for measurements of LV mass [10, 11]. CT was chosen for this study because of increased availability at our institution. It is less commonly used for the assessment of LV mass, but has been validated [13, 14], and does not rely on the geometric assumptions inherent in echocardiographic assessments. The study is also limited by small numbers and is underpowered to detect any sex-related findings.

The initial inclusion criteria were patients with a cardiac CT on record; this population may not reflect the overall general population. Patients with technically difficult echocardiographic studies were excluded from the study, which may lead to other biases in findings. Due to incomplete height and weight data, LV mass was not indexed to body size. However as both CT and echo measurements were performed on the same population of subjects, this should not bias the results. The 2D echocardiographic measurements chosen for the study were the measurements of the septum and inferolateral wall, which are obtained for the calculation of LV mass as per the ASE and the European Association of Cardiovascular Imaging. Further analyses using additional LV measurements may provide further insight.

In this study comparing echo and cardiac CT, measurement of mid-septal wall thickness instead of proximal septal thickness significantly improved the correlation in calculations of LV mass in the elderly. This finding or alternative adjustments that correct for the 5–7% decrease in LV length should be evaluated in larger scale studies. This will also allow for appropriate gender analyses, as there are differences in remodeling between women and men. The ratio between the proximal and mid-septal wall thickness, and aortoseptal angulation may provide alternate methods of correcting LV mass.

Based on this study's findings, further understanding of age-related cardiovascular remodeling and how it is reflected on the various imaging modalities, will bring a deeper understanding of its prognostic implications.

In conclusion, the calculated echocardiographic LV mass remains one of the most powerful predictors of cardiovascular prognosis but may not accurately reflect actual myocardial mass.

Echocardiographic LV mass is likely to overestimate LV mass, particularly in older patients, likely due to LV remodeling with aging. Simple adjustments to the calculation may improve the correlation. Analysis of echo, CT and MR changes with aging may help clarify the patterns of remodeling and elucidate predictors of prognosis.

## Supporting information

**S1 Dataset. Echo and CT dataset.** This is the dataset of measurements taken from echo and CT stratified by age and gender.
(XLS)

**S1 Fig. Angulation measures on echo and CT.** A. Echo Parasternal long axis view: Left ventricular outflow tract angulation (133.5) and aortoseptal angulation (127.1). B. CT three chamber view–left ventricular internal dimension (green), left ventricular outflow tract angle (blue), aortoseptal angle (red)
(TIF)

**S2 Fig. CT measurement of LV mass.** CT four chamber view–left ventricular internal dimension (green), septal wall thickness (blue), posterior wall thickness (yellow), LV length (red)
(TIF)

## Author Contributions

**Conceptualization:** Joshua Stokar, David Leibowitz, Ronen Durst, Dorith Shaham, Donna R. Zwas.

**Data curation:** Joshua Stokar, David Leibowitz, Ronen Durst, Dorith Shaham, Donna R. Zwas.

**Formal analysis:** Joshua Stokar, Ronen Durst, Donna R. Zwas.

**Investigation:** Joshua Stokar, David Leibowitz, Ronen Durst, Donna R. Zwas.

**Methodology:** Joshua Stokar, David Leibowitz, Ronen Durst, Donna R. Zwas.

**Project administration:** Ronen Durst, Dorith Shaham, Donna R. Zwas.

**Resources:** David Leibowitz, Ronen Durst, Donna R. Zwas.

**Software:** Ronen Durst.

**Supervision:** David Leibowitz, Ronen Durst, Dorith Shaham, Donna R. Zwas.

**Validation:** Ronen Durst, Donna R. Zwas.

**Writing – original draft:** Joshua Stokar, David Leibowitz, Ronen Durst, Donna R. Zwas.

**Writing – review & editing:** Joshua Stokar, David Leibowitz, Ronen Durst, Dorith Shaham, Donna R. Zwas.

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
