## [Decision Letter · Decision Letter 0]

11 Jul 2019

PONE-D-19-15733

Echocardiography Overestimates LV Mass in the Elderly as Compared to Cardiac CT

PLOS ONE

Dear Dr. Zwas,

Thank you for submitting your manuscript to PLOS ONE. After careful consideration, we feel that it has merit but does not fully meet PLOS ONE’s publication criteria as it currently stands. Therefore, we invite you to submit a revised version of the manuscript that addresses the points raised during the review process.

We would appreciate receiving your revised manuscript by Aug 25 2019 11:59PM. To enhance the reproducibility of your results, we recommend that if applicable you deposit your laboratory protocols in protocols.io, where a protocol can be assigned its own identifier (DOI) such that it can be cited independently in the future. For instructions see: http://journals.plos.org/plosone/s/submission-guidelines#loc-laboratory-protocols

We look forward to receiving your revised manuscript.

Kind regards,

Giacomo Pucci

Academic Editor

PLOS ONE

**Journal Requirements:**

2. In ethics statement in the manuscript and in the online submission form, please provide additional information about the patient records used in your retrospective study. Specifically, please ensure that you have discussed whether all data were fully anonymized before you accessed them and/or whether the IRB or ethics committee waived the requirement for informed consent. If patients provided informed written consent to have data from their medical records used in research, please include this information.

**Comments to the Author**

1. Is the manuscript technically sound, and do the data support the conclusions?

Reviewer #1: No

Reviewer #2: Yes

2. Has the statistical analysis been performed appropriately and rigorously? 

Reviewer #1: Yes

Reviewer #2: Yes

3. Have the authors made all data underlying the findings in their manuscript fully available?

Reviewer #1: No

Reviewer #2: Yes

4. Is the manuscript presented in an intelligible fashion and written in standard English?

Reviewer #1: Yes

Reviewer #2: Yes

5. Review Comments to the Author

Reviewer #1: The main findings of this study is to correlate measurement of LV mass made by CT with traditional measurement made with echocardiography. Authors demonstrated that there is a good correlation with the 2 way of measurement but in elderly the echo systematically overestimate LV mass.

The main problem with the study is related to the methodological and related to the wrong way of measurement of LV septal wall thickness. As they report in figure 1 they both use proximal and and mid septum measurements and when using mid septum measurement much of the echo-CT discrepancy is eliminated. But this is just because using the proximal measurement in old patients with septal bulge is just uncorrected and the main finding is justified by the high prevalence of septal bulge in old patients. This is a main issue to be solved.

Minor Issue:

- the characteristics of the study population is not described. Are all patients hypertensive? what was the reason of the CT scan? what was the prevalence of hypertension or aortic valve disease in the two groups? Have you excluded patients with severe aortic stenosis?

- in the results section, please report the median and IQR of the time between CT and Echo. It seems that in some patients the echo and CT have been within more than 3 months

-in the discuss section there is a long paragraph related to the fibrosis and increase cardiac collagen content. this is not pertinent to the this study.

Reference: Please update

Reviewer #2: In the article entitled "Echocardiography Overestimates LV Mass in the Elderly as Compared to Cardiac CT" the authors present the results of a cohort of 129 patients which had undergone both echocardiography and cardiac CT as part of their clinical workup. Analysis consisted of comparing LV geometric measurement between modalities in two cohorts of patients: younger than 65 yo and older. The main conclusion of this work is that LV mass correlates well between modalities, but agreement is better in young populations than in the older population. The authors convincingly argue that overestimation of echo-derived LV mass in the elderly is due to inaccurate assumptions of LV remodeling with age and briefly propose alternative methods to increase agreement between modalities.

The article is well written and easy to follow. Study design, data analysis and statistical analysis are all appropriate. Nevertheless several important improvements should be made to the manuscript to help with readability and completeness:

1) In general the tables should be reformatted to avoid character spillover between lines.

2) Figures 3, 4 and 5, need to be reworked to make them more legible (larger font size), include measurement units on axis titles, and appropriate number of decimal places (Figure 3 in particular)

3) Throughout the document measurement units are absent in several places (e.g. LV mass in g) or erroneous (e.g. Table 1 RWT is unit-less, not cm)

4) Table 2 - should include LV mass data as this is the main metric discussed in the study.

5) There is mention of LV volumes briefly in the Discussion section, but these data are not shown. Consider adding the data to the tables and analysis if available or explaining its absence in more detail.

6) Readers may be interested in having measures LV mass correlation between modalities (for each cohort separately and combined). In addition, the linear regression function may be of interest as a method to calibrate between the two modalities. These data should be easily derived from the analysis in Figure 3.

7) On the Bland-Altman plots - please add the p-value for mean difference (bias) different from 0? These are presented in the text and should correspond.

8) The text refers to Figure 4 as an appendix - I believe this is a typo.

9) Some readers may benefit of an explanation of why you analyzed only the septum and posterior wall thicknesses (i.e. why are other walls excluded).

10) The discussion mentions several adjustments to echo LV mass measurements to improve agreement with CT and CMR, but falls short of making firm recommendations. Should echo LV mass be calibrated with an age adjusted function? should mid septum measurements be used? Or perhaps this data is too preliminary to make a firm recommendation?

6. PLOS authors have the option to publish the peer review history of their article (what does this mean?). If published, this will include your full peer review and any attached files.

Reviewer #1: No

Reviewer #2: No

---

## [Author Response · Author response to Decision Letter 0]

11 Aug 2019

11 August 2019

Joerg Heber

Editor-in-Chief, PLOS ONE – Public Library of Science

Dear Prof. Heber:

We very much appreciate the opportunity to revise and resubmit our paper 

“Echocardiography Overestimates LV Mass in the Elderly as Compared to Cardiac CT”

PONE-D-19-15733

to PLOS ONE. We have addressed the concerns of the academic editor and reviewers, as described below.

Academic Editor:

Authors’ response: We have edited the manuscript to meet PLOS ONE’s style requirements.

2. In ethics statement in the manuscript and in the online submission form, please provide additional information about the patient records used in your retrospective study. Specifically, please ensure that you have discussed whether all data were fully anonymized before you accessed them and/or whether the IRB or ethics committee waived the requirement for informed consent. If patients provided informed written consent to have data from their medical records used in research, please include this information.

Authors’ response: We have added the IRB ethics committee number and added the statement:

“The ethics committee waived the requirement for informed consent.”

Response to Reviewers

Reviewer #1:

 The main findings of this study is to correlate measurement of LV mass made by CT with traditional measurement made with echocardiography. Authors demonstrated that there is a good correlation with the 2 way of measurement but in elderly the echo systematically overestimate LV mass.

The main problem with the study is related to the methodological and related to the wrong way of measurement of LV septal wall thickness. As they report in figure 1 they both use proximal and and mid septum measurements and when using mid septum measurement much of the echo-CT discrepancy is eliminated. But this is just because using the proximal measurement in old patients with septal bulge is just uncorrected and the main finding is justified by the high prevalence of septal bulge in old patients. This is a main issue to be solved.

Authors’ response: The authors agree with the reviewers that measurements performed according to present guidelines necessitate the inclusion of the septal bulge which leads to overestimation of LV mass. This study seeks to address precisely the concern raised by the reviewer.

Minor Issues:

- the characteristics of the study population is not described. Are all patients hypertensive? what was the reason of the CT scan? what was the prevalence of hypertension or aortic valve disease in the two groups? Have you excluded patients with severe aortic stenosis?

Authors’ response: As this was a retrospective study, the indications for imaging were not available to the investigators. Patients with severe aortic stenosis were included. The investigators believe that this broadens the spectrum of ventricular hypertrophy seen in the study, making the findings more robust and reflecting clinically relevant situations where estimation of LV mass is of importance. 

- in the results section, please report the median and IQR of the time between CT and Echo. It seems that in some patients the echo and CT have been within more than 3 months

Authors’ response: The authors appreciate this concern. The manuscript was updated with the correct information in the first paragraph of the results section. The sentence in the results section now reads: 

“ Mean number of days between CT and echo studies was 3 with IQR of 34 days. “

-in the discussion section there is a long paragraph related to the fibrosis and increase cardiac collagen content. this is not pertinent to the this study. 

Authors’ response:The authors respect the reviewers comment. The two sentences discussing fibrosis, and myocardial stiffness were removed from the manuscript.

Reviewer #2: 

1) In general the tables should be reformatted to avoid character spillover between lines.

Authors’ response: The tables have been updated so as to avoid character spillover between lines.

2) Figures 3, 4 and 5, need to be reworked to make them more legible (larger font size), include measurement units on axis titles, and appropriate number of decimal places (Figure 3 in particular)

Authors’ response: Figures 3,4, and 5 have been reworked with larger font sizes, measurement units and decimal places, as suggested by the reviewer. 

3) Throughout the document measurement units are absent in several places (e.g. LV mass in g) or erroneous (e.g. Table 1 RWT is unit-less, not cm)

Authors’ response: The authors appreciate the reviewer’s comment and have placed the measurement units in the appropriate places in the document. 

4) Table 2 - should include LV mass data as this is the main metric discussed in the study.

Authors’ response: The authors agree that the LV mass data is the main metric in the document. We have, however, placed this data in Table 3, as it permits a clearer and more informative presentation of the core findings of the study. 

5) There is mention of LV volumes briefly in the Discussion section, but these data are not shown. Consider adding the data to the tables and analysis if available or explaining its absence in more detail.

Author’s response: The authors appreciate this comment as it is an important point. The data have been added to the table, and comments have been added in the results section. The addition reads: 

“There was no significant difference between LV volumes as calculated by echocardiography and CT.”

and

“No differences between age groups in either modality were seen in left ventricular internal diameter, volume or sphericity index.”

6) Readers may be interested in having measures LV mass correlation between modalities (for each cohort separately and combined). In addition, the linear regression function may be of interest as a method to calibrate between the two modalities. These data should be easily derived from the analysis in Figure 3.

Authors’ response: This comment is well taken. We have incorporated the reviewer’s comment into the results section and Figure 3 which now reads:

“Relative correlation coefficient between ASE and CT derived LV mass was 0.89; correlations by age group are presented in the scatter plot in figure 3. “ 

7) On the Bland-Altman plots - please add the p-value for mean difference (bias) different from 0? These are presented in the text and should correspond.

Authors’ response: As the reviewer suggested ,this was incorporated into Figure 4. 

8) The text refers to Figure 4 as an appendix - I believe this is a typo.

Authors’ response; This was a typo and has been corrected. 

9) Some readers may benefit of an explanation of why you analyzed only the septum and posterior wall thicknesses (i.e. why are other walls excluded). 

Authors’ response: The authors chose to use the measurements that are standardly obtained in echocardiography laboratories and are required for calculation of LV mass according to ASE and European Association of Cardiovascular Imaging. . We added this valid point to the study limitations. 

10) The discussion mentions several adjustments to echo LV mass measurements to improve agreement with CT and CMR, but falls short of making firm recommendations. Should echo LV mass be calibrated with an age adjusted function? should mid septum measurements be used? Or perhaps this data is too preliminary to make a firm recommendation? 

Authors’ response. The authors appreciate the reviewer’s acceptance of our findings. We believe that a larger follow up studies is needed to confirm these findings before making a firm recommendation for updating the guidelines to use mid-septal measurements. 

We appreciate the interest that you and the reviewers have taken in this manuscript.

---

## [Decision Letter · Decision Letter 1]

13 Sep 2019

PONE-D-19-15733R1

Echocardiography Overestimates LV Mass in the Elderly as Compared to Cardiac CT

PLOS ONE

Dear Dr. Zwas,

Thank you for submitting your manuscript to PLOS ONE. After careful consideration, we feel that it has merit but does not fully meet PLOS ONE’s publication criteria as it currently stands. Therefore, we invite you to submit a revised version of the manuscript that addresses the points raised during the review process

We would appreciate receiving your revised manuscript by Oct 28 2019 11:59PM. To enhance the reproducibility of your results, we recommend that if applicable you deposit your laboratory protocols in protocols.io, where a protocol can be assigned its own identifier (DOI) such that it can be cited independently in the future. For instructions see: http://journals.plos.org/plosone/s/submission-guidelines#loc-laboratory-protocols

We look forward to receiving your revised manuscript.

Kind regards,

Giacomo Pucci

Academic Editor

PLOS ONE

Reviewers' comments:

Reviewer's Responses to Questions

**Comments to the Author**

1. If the authors have adequately addressed your comments raised in a previous round of review and you feel that this manuscript is now acceptable for publication, you may indicate that here to bypass the “Comments to the Author” section, enter your conflict of interest statement in the “Confidential to Editor” section, and submit your "Accept" recommendation.

Reviewer #1: (No Response)

Reviewer #2: All comments have been addressed

2. Is the manuscript technically sound, and do the data support the conclusions?

Reviewer #1: Yes

Reviewer #2: Yes

3. Has the statistical analysis been performed appropriately and rigorously? 

Reviewer #1: Yes

Reviewer #2: Yes

4. Have the authors made all data underlying the findings in their manuscript fully available?

Reviewer #1: Yes

Reviewer #2: Yes

5. Is the manuscript presented in an intelligible fashion and written in standard English?

Reviewer #1: Yes

Reviewer #2: Yes

6. Review Comments to the Author

Reviewer #1: The comments have been correctly answered by the Authors, but the discuss section needs to be better focused on the main concern raised by the reviewer. IF the main point is to address the issue of incorrect measurement of LV mass in older people with septal bulge, this should be better underlined.

Regarding the days between the ECHO and CT, please report median and not mean.

Reviewer #2: The authors have generally addressed this reviewers’ comments, however few corrections are still required:

1) Page 9 - reference "{Lang, 2015 #34}" should be corrected.

2) Page 9 - after "(folded empirical cumulative distribution plots" a bracket is missing.

3) Page 10 - comma missing after "IBM".

4) Page 10 - Results - mean number of days between CT and echo was previously reported as 19 and now as 3. What is the reason for this change? Is there a typo and mean should be median???

5) Figure 3 caption - the line of identity is descried as black but, on my screen, appears blue (might be low quality rendering). Please ensure consistency.

6) Figure 4 caption - refers to multiple Bland-Altman plots, but only 1 was included in this revision (original submission had 2).

7) Page 19 there is an erroneous line break before references [29, 30].

7. PLOS authors have the option to publish the peer review history of their article (what does this mean?). If published, this will include your full peer review and any attached files.

Reviewer #1: Yes: Costantino Mancusi

Reviewer #2: Yes: Ran Klein PhD, University of Ottawa

---

## [Author Response · Author response to Decision Letter 1]

24 Sep 2019

Dear Prof. Heber:

We very much appreciate the opportunity to revise and resubmit our paper 

“Echocardiography Overestimates LV Mass in the Elderly as Compared to Cardiac CT”

PONE-D-19-15733

to PLOS ONE. We have addressed the concerns of the reviewers, as described below.

Reviewer #1: The comments have been correctly answered by the Authors, but the discuss section needs to be better focused on the main concern raised by the reviewer. IF the main point is to address the issue of incorrect measurement of LV mass in older people with septal bulge, this should be better underlined.

Authors response: Two sections have been added to the discussion section to address the concern of the reviewer.

The first on page 17 now reads:

“This buckling phenomenon, known as discrete upper septal hypertrophy, also known as a septal bulge, has been well-described in the elderly, and is found in up to 10% of large echocardiographic cohorts.(26, 27) This finding is associated with hypertension and diastolic dysfunction, but not with increased cardiovascular mortality.(28) It has been associated with increased LV mass in echocardiographic studies.(29)”

The second on page 18 now reads:

“As the proximal septal thickening is not necessarily focal but may have a somewhat gradual attenuation, this study suggests that even measurements immediately distal to the septal bulge may lead to overestimation of LV mass. Although mid-ventricular measurement did not specifically adjust for the geometric changes, the estimate of LV mass correlated better with the 3-dimensional assessment performed by CT. This study suggests that mid-septal rather than standard proximal septal measurement may increase the accuracy of echocardiographic assessment of LV mass, as the septal bulge reflects remodeling and compression rather than an increase in overall ventricular mass. “

Regarding the days between the ECHO and CT, please report median and not mean.

Authors response: The manuscript now reports the median and not the mean.

Reviewer #2: The authors have generally addressed this reviewers’ comments, 

however few corrections are still required:

1) Page 9 - reference "{Lang, 2015 #34}" should be corrected.

Corrected in manuscript.

2) Page 9 - after "(folded empirical cumulative distribution plots" a bracket is missing.

Corrected in manuscript.

3) Page 10 - comma missing after "IBM".

Corrected in manuscript.

4) Page 10 - Results - mean number of days between CT and echo was previously reported as 19 and now as 3. What is the reason for this change? Is there a typo and mean should be median???

As pointed out by the reviewer, it should be median. Corrected in manuscript.

5) Figure 3 caption - the line of identity is descried as black but, on my screen, appears blue (might be low quality rendering). Please ensure consistency.

Corrected in manuscript.

6) Figure 4 caption - refers to multiple Bland-Altman plots, but only 1 was included in this revision (original submission had 2).

The second Bland-Altman was by mistake not uploaded in the initial revision, and will be uploaded now.

7) Page 19 there is an erroneous line break before references [29, 30].

Corrected in manuscript.

Thanks very much.

---

## [Decision Letter · Decision Letter 2]

7 Oct 2019

Echocardiography Overestimates LV Mass in the Elderly as Compared to Cardiac CT

PONE-D-19-15733R2

Dear Dr. Zwas,

We are pleased to inform you that your manuscript has been judged scientifically suitable for publication and will be formally accepted for publication once it complies with all outstanding technical requirements.

With kind regards,

Giacomo Pucci

Academic Editor

PLOS ONE

Additional Editor Comments (optional):

Reviewers' comments:

Reviewer's Responses to Questions

**Comments to the Author**

1. If the authors have adequately addressed your comments raised in a previous round of review and you feel that this manuscript is now acceptable for publication, you may indicate that here to bypass the “Comments to the Author” section, enter your conflict of interest statement in the “Confidential to Editor” section, and submit your "Accept" recommendation.

Reviewer #1: All comments have been addressed

2. Is the manuscript technically sound, and do the data support the conclusions?

Reviewer #1: Yes

3. Has the statistical analysis been performed appropriately and rigorously? 

Reviewer #1: Yes

4. Have the authors made all data underlying the findings in their manuscript fully available?

Reviewer #1: (No Response)

5. Is the manuscript presented in an intelligible fashion and written in standard English?

Reviewer #1: Yes

6. Review Comments to the Author

Reviewer #1: (No Response)

7. PLOS authors have the option to publish the peer review history of their article (what does this mean?). If published, this will include your full peer review and any attached files.

Reviewer #1: Yes: Costantino Mancusi

---

## [Editor Report · Acceptance letter]

17 Oct 2019

PONE-D-19-15733R2 

Echocardiography Overestimates LV Mass in the Elderly as Compared to Cardiac CT 

Dear Dr. Zwas:

I am pleased to inform you that your manuscript has been deemed suitable for publication in PLOS ONE. Congratulations! Your manuscript is now with our production department. 

With kind regards,

on behalf of

Dr. Giacomo Pucci 

Academic Editor

PLOS ONE